# *dCas9-BE*3 and *dCas12a-BE3* Systems Mediated Base Editing in Kiwifruit Canker Causal Agent *Pseudomonas syringae* pv. *actinidiae*

**DOI:** 10.3390/ijms24054597

**Published:** 2023-02-27

**Authors:** Bo Liu, Wenpeng Song, Linchao Wang, Yantao Wu, Xiaoting Xu, Xiangli Niu, Shengxiong Huang, Yongsheng Liu, Wei Tang

**Affiliations:** 1Ministry of Education Key Laboratory for Bio-Resource and Eco-Environment, College of Life Science, Sichuan University, Chengdu 610064, China; 2Sichuan Provincial Key Laboratory for Development and Utilization of Characteristic Horticultural Biological Resources, College of Chemistry and Life Sciences, Chengdu Normal University, Chengdu 611130, China; 3School of Horticulture, Anhui Agricultural University, Hefei 230036, China; 4School of Food and Biological Engineering, Hefei University of Technology, Hefei 230601, China

**Keywords:** *Pseudomonas syringae* pv. *actinidiae*, base editing, dCas9-BE3, dCas12a-BE3, multi-site knockout, mutant library

## Abstract

*Pseudomonas syringae* pv. *actinidiae* (*Psa*) causes bacterial canker of kiwifruit with heavy economic losses. However, little is known about the pathogenic genes of *Psa*. CRISPR (Clustered Regularly Interspaced Short Palindromic Repeats)/Cas-mediated genome editing technology has dramatically facilitated the characterization of gene function in various organisms. However, CRISPR genome editing could not be efficiently employed in *Psa* due to lacking homologous recombination repair. The base editor (BE) system, which depends on CRISPR/Cas, directly induces single nucleoside C to T without homology recombination repair. Here, we used dCas9-BE3 and dCas12a-BE3 systems to create substitutions of C to T and to convert CAG/CAA/CGA codons to stop codons (TAG/TAA/TGA) in *Psa*. The dCas9-BE3 system-induced single C-to-T conversion frequency of 3 to 10 base positions ranged from 0% to 100%, with a mean of 77%. The dCas12a-BE3 system-induced single C-to-T conversion frequency of 8 to 14 base positions in the spacer region ranged from 0% to 100%, with a mean of 76%. In addition, a relatively saturated *Psa* gene knockout system covering more than 95% of genes was developed based on dCas9-BE3 and dCas12a-BE3, which could knock out two or three genes at the same time in the *Psa* genome. We also found that *hopF2* and *hopAO2* were involved in the *Psa* virulence of kiwifruit. The HopF2 effector can potentially interact with proteins such as RIN, MKK5, and BAK1, while the HopAO2 effector can potentially interact with the EFR protein to reduce the host’s immune response. In conclusion, for the first time, we established a PSA.AH.01 gene knockout library that may promote research on elucidating the gene function and pathogenesis of *Psa*.

## 1. Introduction

CRISPR (Clustered Regularly Interspaced Short Palindromic Repeats)/Cas proteins constitute a prokaryotic immune surveillance system in bacteria and archaea to defend against the infection of phages [1,2,3]. Recently, the CRISPR/Cas9 system has been engineered and widely used as a genome editing tool [4]. The CRISPR-Cas9 editing system consists of the Cas9 protein and sgRNA. The sgRNA enables Cas9 to recognize and cleave specific target DNA sequences, resulting in double-strand breaks (DSBs). The recognition of target DNA requires the presence of a short protospacer adjacent motif (PAM) flanking the target DNA with a structure of ‘NGG’ in the CRISPR/Cas9 system [5]. Alternatively, CRISPR/Cas12a (cpf1) is derived from another adaptive immune system of bacteria. Similar to CRISPR/Cas9, it can recognize and cut targeted DNA sequences, producing double-stranded DNA breaks [6,7]. The crRNAs (CRISPR RNAs) guide the Cas12a nucleases to cleave both DNA strands with a single RuvC-like nuclease domain. Cas12a can cut DNA in a staggered orientation within PAM (TTTV)-distal regions of the protospacer [8,9,10]. The non-homologous end joining (NHEJ) mechanism sets a foundation in a significant repair pathway for DNA double-stranded breaks (DSBs) that is prone to insertion and/or deletion mutations at the junctional site and generates a frameshift mutation that disrupts the targeted gene in eukaryotes [11].

Base editing is a new powerful technology that enables the programmable conversion of single nucleosides in the eukaryote and prokaryote genomes [12]. Base editors are usually fused with a defective Cas9 protein (Cas9D10A or Cas9D10AH840A) and a deaminase or combined with a defective Cas12a (dLbCpf1D832A) protein and a deaminase. Guided by the dCas9/sgRNA complex or the dCpf1/crRNA complex, the deaminase can be recruited to the target site to perform base editing [9]. The commonly used base editors include CBEs (cytosine base editor) and ABEs (adenine base editor), typically generating C·G-to-T·A or A·T-to-G·C conversions, respectively [13]. The base editing system avoids DSBs and improves the efficiency of genome editing without random mutations compared to non-homologous end joining repair [14].

Bacterial canker severely threatens kiwifruit cultivation worldwide [15]. *Pseudomonas syringae* pv. *actinidia*e (*Psa*) is the causal agent of bacterial canker, primarily infecting commercial cultivars of *Actinidia deliciosa* and *A. chinensis* [16]. *Psa* infects the kiwifruit plant through stomata, stylar, and wounds [17]. *Psa*-infected plant displays only minor symptoms in summer, while in winter, with humid conditions, *Psa* colonizes various kiwifruit tissues and rapidly becomes systemic, leading to the death of the host plant [15]. Although *Psa* is a very aggressive pathogen to kiwifruit worldwide, we know little about its pathogenic mechanism. The type III secretion system (T3SS) has been shown to be closely associated with the pathogenicity of bacteria [18]. *Pseudomonas syringae* pv. *tomato* DC3000, which is used as a model organism for studying plant pathogenicity, has already investigated the function of some effector proteins secreted by the T3SS, such as HopF2_Pto_ and HopAO1_Pto_ [19,20]. Since the effector protein of *Psa* has high homology with the DC3000 effector protein, we can refer to the study of DC3000 effector proteins to explore the pathogenic mechanism of *Psa*.

The currently used method for constructing mutations in *Psa* strains is homologous recombination (HR) [21,22,23]. Since bacterial cells do not possess the NHEJ pathway, only the cells that have undergone homologous recombination can survive after the double-stranded DNA break of the genome [24]. The RecBCD or λ-Red recombination system in bacteria needs upstream and downstream homologous arms to mediate recombination events [22]. The obvious disadvantage of these knockout methods in *Psa* is that they are cumbersome. In addition, they are challenging to use for knockout multi-site or high-throughput gene mutation experiments.

Recently, the genome editing tool CRISPR/Cas9 system has been used to create large fragment deletions (7829 bp to 64,412 bp) in the *Psa* genome [25]. Generally, the CRISPR/Cas9 system is employed to assist in editing a bacterial genome, needing donor DNA as an editing template [26]. However, Ho et al. reported that repairing DSBs caused by the CRISPR/Cas 9 system in *Psa* might depend on NHEJ without donor DNA. The applicability of this method may be limited because it generates large fragment deletions, and deletion sites are uncontrollable [25].

Base editing has been demonstrated as a powerful tool with high mutation frequencies in bacteria [24,27,28]. In this work, we used dCas9-BE3 and dCas12a-BE3 systems to induce the substitution of C to T to convert CAG/CAA/CGA codons into a stop codon (TAG/TAA/TGA) in *Psa*. We showed that BE3 achieved accurate and effective base conversion in *Psa*. We also demonstrated that BE3 could target multi-site base conversion by a single plasmid. Our purpose is to provide a new, simple, and high-throughput tool to investigate the function of *Psa* genes.

## 2. Result

### 2.1. Genome Information of PSA.AH.01

The genomic DNAs of strain PSA.AH.01 (PRJNA923731), isolated from Sichuan Province in China, were sequenced by Oxford Nanopore Platform (Oxford Nanopore, Oxford, UK). A circular chromosome and a circular plasmid were obtained by genome de novo assembly. The length of the chromosome was 6,529,123 bp, and the length of the plasmid was 77,260 bp, for a total of 6,606,383 bp. The GC content of the chromosome was 58.42%, and the plasmid was 56.00%. The assembled data of PSA.AH.01 were subjected to Glimmer, version 3.02, and 6079 protein-coding sequences (CDSs). Furthermore, 65 tRNA genes, 16 rRNA genes, and 43 sRNA genes were identified. Subsequently, PSA.AH.01 was annotated using the Bacterial Virulence Factor Database [29]. The results showed that it annotated 552 genes, including 31 effector proteins of the T3SS (HopAS1, HopAN1, HopAO2, HopAC1, HopW1, HopR1, HopAG1, HopAH1, HopAI1, HopAK1, AvrPto, HopAH2-1, HopAH2-2, HopAE1, AvrRpm1, AvrE1, HopM1, HopAA1-1, HopN1, HopJ1, HopI1, HopAJ2, AvrB4, HopD1, HopQ1, HopA1, HopZ3, HopAV1, HopAU1, HopF2, and HopAA1-2).

### 2.2. Successfully Knockout Flic in Psa

The dCas9-BE3 system could convert CAG/CAA (Gln) or CGA (Arg) codons into respective TAG/TAA/TGA premature Stop codons. To test the efficiency of the dCas9-BE3 system in *Psa*, we constructed a dCas9-BE3-*fliC* plasmid vector that harbors gRNA to target the flagellin (*fliC*) gene in *Psa*. The sgRNAs were designed between the 382 bp and 401 bp of the *fliC* open reading frame to truncate the protein (Figure 1A). The dCas9-BE3-*fliC* plasmid was successfully transferred into *Psa* chemically competent cells. Twenty independent clones were randomly selected, and the *fliC* gene of all clones was amplified and sequenced using site-specific primers. Sanger sequencing results showed that dCas9-BE3-mediated base editing in *Psa* was almost 100% (Figure 1B). Scanning electron microscopy was performed to observe the flagellin of *Psa* and Δ*fliC*. Several flagella in the wild type were observed, while no flagella in the Δ*fliC* were detected (Figure 1C,D). The result shows that the flic protein has been successfully truncated.

### 2.3. Editing Rule of dCas9-BE3 in Psa

To evaluate its general efficacy in *Psa*, we tested dCas9-BE3 at twelve target gene sites with the NGG PAM sequence (Figure 2A). These genes are putatively related to *Psa* pathogenicity, including *avrB4*, *hopR1*, *hrcC*, *hopAO2*, *hopF2*, *hopM1*, *hopZ3*, *hopAI1*, *hopQ1,* and *hopI1*. Each target gene sequence from ten positive clones was amplified and sequenced. Among the 12 target sites with the NGG PAM sequence, dCas9-BE3-induced base editing efficiency was almost 100% in *Psa* (Figure 2B). The single C-to-T conversion frequency of the 3rd to 10th base position ranged from 0% to 100% with a mean of 77%. Assuming the cytosines were at 3< or >10, the dCas9-BE3-induced mutation efficiency was relatively low (from 0% to 80%, with a mean of 0.04%). The editing box of dCas9-BE3 in *Psa* was from the 3rd to 10th base positions (Figure 2C). However, compared with the TC/CC/AC structure in the editing box, the base editing efficiency of the GC structure was low (Figure 2B). This result is in agreement with findings from previous studies showing that APOBEC1 was not able to act efficiently on the cytosines following guanosine deaminate cytosines with a 5′-guanosine.

### 2.4. Multi-Site Knockout by Using the dCas9-BE3 System

Homologous recombination is the traditional gene knockout method in bacteria, but it requires much work when multiple genes need to be knockout simultaneously. CRISPR-Cas9 had an inherent advantage in the multi-site knockout. Therefore, the experiment attempted to construct multiple target sequences in the dCas9-BE3 vector to study the possibility of simultaneously knocking out two or three genes.

The dCas9-*BE3*-*HopAI1-HopI1, dCas9-BE3-AvrE-HopR1,* and *dCas9-BE3-AvrE-HopM1-HopR1* vectors were constructed and transformed into *Psa* (Appendix A and Supplementary Notes 3). The results showed that the 15th amino acid *Gln* (CAG) in *hopAI1* and the 177th amino acid *Gln* (CAG) in *hopI1* were successfully mutated to stop codons (TAG). The sequencing of *avrE1* and *hopR1* in positive colonies showed that the CAG or CAA (Gln) codons were successfully converted to a TAG or TAA stop codon (Appendix A). Similarly, the *avrE1*, *hopM1*, and *hopR1* genes were also successfully converted into stop codons in the same colony (Figure 3). The C to T-conversion frequency was 100% at the editing window. The results showed that when using dCas9-BE3 in *Psa*, the efficiency of the multi-site knockout was as high as that of the single-site knockout strategy.

### 2.5. pdCas12a-BE Knockout System

Although the dCas9-BE3 system was effective in *Psa*, its application was limited to the G/C-rich protospacer, PAM, editing window, and the position of CAA, CAG, and CGA codons. We developed a dCas12a-BE editing system to overcome this limitation to recognize a T-rich PAM and catalyze C-to-T conversion in *Psa*. The dCas12a sequence was amplified and replaced the dCas9 region in the dCas9-BE3 to construct pdCas12a-BE (Appendix A). To evaluate its general efficacy in *Psa*, we tested pdCas12a-BE at six target gene sites, such as *hopAI1*, *hopI1*, *hopQ1*, and *Flic* loci (Figure 4A). Among the six target sites with the TTTV PAM sequence, dCas12a-BE3-induced base editing ranged from 70% to 100% efficiency in *Psa*. The single C-to-T conversion frequency of 8 to 14 bases in the spacer region ranged from 0% to 100% with a mean of 76% (Figure 4B). If the cytosines were at 8< or >14 in the spacer region, the dCas12a-BE3 editing efficiency was relatively low (from 0% to 100% with a mean of 0.04%). This result indicated that the major editing window ranges from positions 8 to 14 in the spacer region (Figure 4C). Sometimes, the TC structure out of the editing window could also be converted to TT.

### 2.6. Construction of sgRNA Libraries Targeting Genome-Wide Editing

In order to design highly specific sgRNAs and generate loss-of-function mutations efficiently, we used Python to screen out targets in the PSA.AH.01 genome (Supplementary Notes 4). In total, 9467 target sites were identified and located in 5728 genes. The two 25-bp homologous arms derived from the dCas9-BE3 were added to the two ends of the 20-bp guide sequences. Furthermore, 9467 oligonucleotides were synthesized by array-based oligo-pool synthesis. Then the oligonucleotides were amplified, purified, and ligated into the pBmBE3 vector. The recombinant plasmids were transformed into *E. coli* DH5α chemically competent cells. More than 1.7 × 10^7^ clones were grown on the selection plates (Gm). The number of plasmid DNAs was approximately 1657 times of library size. All clones were harvested and combined to prepare the plasmid DNAs constituting the sgRNA library. The primers BE3F/R were used to amplify the plasmid DNA library and PCR products were deeply sequenced by Next-generation sequencing (HiSeq 2500). The sequence data showed that 9464 of the 9467 sgRNAs were represented by at least one read (99.97%), and these 9464 sgRNAs covered 5725 genes (99.9%). Moreover, 90% of the sgRNAs had approximately 1721 reads, and 10% of the sgRNAs had less than 460 reads. We randomly selected 365 clones and sequenced them. The results showed that 307 of the 365 clones harbored the correct sgRNA target sequences (84.1%), 55 of the 365 clones had mutated sgRNA target sequences (15.1%), and three clones did not harbor the target sequence (0.8%) (Appendix A). Those results showed that the constructed sgRNAs library is highly qualified with saturated gene coverage and sgRNA accuracy.

### 2.7. Pathogenicity Assays of Psa Mutant Strains

To test the pathogenicity of *Psa* mutant strains constructed by BE3 systems, *‘Hongyang’* tissue culture plantlets were incubated with Δ*hrcC*, Δ*hopR1,* Δ*hopF2,* and Δ*hopAO2* mutant strains. After 20 days, *‘Hongyang’* plantlets inoculated with *Psa3* developed leaf yellowing and browning. The plantlets inoculated with the *ΔhrcC* mutant (the type III secretion system) appeared healthy and asymptomatic (Figure 5A). The symptoms of ‘*Hongyang’* plantlets infected by Δ*hopR1* were lighter than those infected with *Psa*. The pathogenicity of Δ*hrcC* and Δ*hopR1* was similar to those gene mutants constructed by homologous recombination [21]. In addition, we knocked out the *hopF2* and *hopAO2,* and the virulence of Δ*hopF2* and Δ*hopAO2* decreased. The mean leaf yellowing and browning rate of *Psa*, Δ*hrcC*, Δ*hopR1,* Δ*hopF2,* and Δ*hopAO2* was 67%, 10%, 14%, 29%, and 43%, respectively (Figure 5B). The *Psa*, Δ*hrcC*, Δ*hopR1,* Δ*hopF2,* and Δ*hopAO2* had mean bacterial biomass of 5.80, 4.76, 4.94, 5.04, and 5.40 Log10 CFU/cm^2^, respectively (Figure 5C).

## 3. Discussion

The homologous recombination system used to construct mutants in *Psa* usually requires recombinase and homologous fragments [21,22,23]. The process of homologous recombination was complicated, and it needed to be more suitable for multi-site mutations or high-throughput gene knockout strategy. CRISPR had the advantage of easy operation and could cause multi-site knockout in eukaryotes [30]. In our research, the editing window of dCas9-BE3 in *Psa* was from positions 3 to 10 nt. The average editing efficiency was 77%, and the editing window of dCas12a-BE3 was from positions 8 to 14 nt, in which the average editing efficiency was 76%. The BE3 system was first shown to have high efficiency in *Psa*. We also formulated a multi-site knockout system that could knock out two or three genes simultaneously in the *Psa* genome. Through the above method, we found that *hopF2* and *hopAO2* were involved in the *Psa* virulence of kiwifruit.

Ho et al. used the CRISPR/Cas9 system to knock out the non-ribosomal peptide synthetase, resulting in large fragment deletions [25]. The likely reason may be that the non-homologous end-joining repair pathway was incomplete in *Psa*. The BE3 system was successfully used in *E. coli., B. melitensis, P. aeruginosa, P. putida KT2440, P. fluorescens GcM5-1A*, and *P. syringae DC3000* [24]. We used dCas9-BE3 and dCas12a-BE3 systems to produce stop codons to knockout genes without double-strand breaks. The results showed that all positive *Psa* clones were mutated with an efficiency of 100%. Komor et al. constructed the first dCas9-BE3 system, and the editing efficiency was 36–75% in the mammalian cells, of which the editing window was from positions 4 to 8 nt [14]. Li X et al. constructed the dCas12a-BE3 system, and the average editing efficiency was 20% in eukaryotic cells, of which the editing window was from positions 8 to 13 nt [31].

Compared with the dCas12a-BE3 that recognized AT-rich (TTTN) PAM, we thought that the dCas9-BE3, which recognized GC-rich (NGG) PAM, was more suitable for gene knockout in *Psa* because the genome of *Psa* was actually GC-rich and the GC content of PSA.AH.01 was 58.39%. Therefore, according to the editing rules, our experiment analyzed the gene knockout targets in the genomes of five *Psa* strains (PSA.AH.01, ICMP9853, ICMP18708, P220, and M228) with the previous studies of dCas9-BE3 and dCas12a-BE3 in *Psa*. The results showed that the proportions of genes with dCas9-BE3 knockout targets in the *Psa* genomes were 94.23%, 95.85%, 96.68%, 96.44%, and 96.36%, respectively, with a mean of 95.91%. The proportions of genes with dCas12a-BE3 knockout targets in the *Psa* genome were 54.07%, 55.35%, 56.92%, 57.57%, and 58.17%, with a mean of 56.42% (Supplementary Notes 5 and Appendix A). Furthermore, the average proportion of genes with dCas12a-BE3 knockout targets without dCas9-BE3 was 17.82%, and the average proportion of genes with dCas12a-BE3 knockout targets with only one dCas9-BE3 knockout target was 25.40% (Appendix A). In conclusion, most genes in *Psa* had dCas9-BE3 knockout targets. When some genes had none or only one dCas9-BE3 knockout target, dCas12a-BE3 could be an alternate option. We showed that the dCas9-BE3 and dCas12a-BE3 systems could cover more than 95% of genes and knock out two or three genes simultaneously in the *Psa* genome. Therefore, our systems could perform highly efficient base editing in *Psa*, which may allow for efficiently manipulating the *Psa* genome at one or multiple sites at once.

With the *Psa* genome sequencing, the challenge in the post-genome era is to study the function of the *Psa* genes systematically. Gene knockout is the most common and effective way to achieve this goal. Generating large-scale mutants is of great value for studying *Psa* functional genes. Traditional methods for producing large-scale mutations include physical, chemical, and biological methods. The biological methods include transposon insertion and homologous recombination. The transposon insertion method is random and inserts only 47% of chromosomal genes into *Psa* [32]. The homologous recombination method is inefficient and 9.3% of genes in *E. coli* were deleted [33]. The CRISPR-based base editing system can mutate the base C at a specific site in the *Psa* genome to a base T, so the CGA/CAA/CAG of the target gene was mutated to a stop codon and finally generated a mutation. It is because of the convenience that it has been successfully applied to eukaryotic and prokaryotic cells. However, the construction of large-scale mutant libraries of *Psa* is rarely reported. Therefore, we constructed a high-throughput *Psa* mutant library based on the BE3 system to provide an experimental basis for studying gene function. Since there is an almost one-to-one relationship between sgRNA and target genes in these BE3 mutants, the disease-associated genes can easily identify with observed phenotypes.

The study found that HopF2_Pto_ in *Pseudomonas syringae* pv. *tomato* DC3000 can bind to the RIN protein of Arabidopsis and not trigger the plant’s ETI response, thereby promoting the growth of DC3000 within the plant [19]. HopF2_Pto_ can also interact with and inactivate the Arabidopsis MKK5 protein, reducing the plant’s defense ability [34]. Additionally, HopF2_Pto_ can interact with BAK1, inhibiting the phosphorylation of the BIK1 protein in Arabidopsis and reducing the non-host immunity of plants [35]. The virulence activity of HopF2_Pto_ in tomato requires its myristoylation site and the catalytic residue of ADP-RT, with the 71st arginine (Arg_71_) and the 175th aspartic acid (Asp_175_) of HopF2_Pto_ being particularly important [34]. Further analysis of HopF2_Pto_ revealed that it has a protein sequence of 204 amino acids, while HopF2_Psa_ in PSA.AH.01 has 205 amino acids with a homology of 57%. The 72nd amino acid of HopF2_Psa_ is also Arg_72_, and the 174th amino acid is Asp_174_, indicating that HopF2_Psa_ may also interact with proteins such as RIN, MKK5, and BAK1 in kiwifruit (Appendix A).

HopAO1_Pto_ is similar to protein tyrosine phosphatases (PTPs) and has PTP activity, which requires a conserved catalytic cysteine residue (Cys_378_) [20]. HopAO1_Pto_ can inhibit the HR response of tobacco, but when Cys_378_ is changed to Ser, this inhibition disappears, highlighting the importance of Cys_378_ in the role of HopAO1_Pto_ [36]. The study also found that HopAO1_Pto_ acts on the tyrosine 836 site of the Arabidopsis EFR protein, reducing its phosphorylation and preventing the downstream immune response [37]. One study reported that HopAO2 is widely distributed in *Pseudomonas syringae*, with DC3000 possessing only HopAO1 and PSA.AH.01 having only HopAO2. HopAO2 has phosphatase activity and can inhibit early defense responses in tobacco, reduce reactive oxygen species, and reduce callose deposition [38]. The conserved domain of PTP is Cx5R [39], the active site of HopAO1_Pto_ in DC3000 is VHC_378_NGGRGR_384_T, and the conserved domain of HopAO2_Psa_ in PSA.AH.01 is IHC_243_GVGQGR_249_T (Appendix A). Based on this information, we infer that the pathogenicity of the HopAO2 protein in PSA.AH.01 may be similar to that of HopAO1, reducing the host’s immune response by acting on the EFR protein of kiwifruit. Finally, there are limited reports on HopR1. Kvitko et al. (2009) reported that HopR1 of DC3000 can inhibit PTI-related callose deposition in *Nicotiana benthamiana*, improving the reproductive ability of DC3000 [40]. Jayaraman et al. (2020) showed that in *Psa*, the *hopR1* is a key gene for bacterial pathogenesis, which is consistent with our results [21]. However, the function of HopR1 needs more research.

## 4. Materials and Methods

### 4.1. Bacterial Strains and Plant Growth Conditions

The *Psa* strain (PSA.AH.01) isolated in China (Sichuan province) from a leaf spot lesion of *A. deliciosa* cv. Hayward was routinely grown in King’s B medium (KB) at 25 °C. The *Escherichia coli* strain (Trans T1) (Transgen Biotech, Beijing, China) was cultivated at 37 °C in Luria–Bertani (LB) or on LB agar plates. Antibiotics and additives were used at the following final concentrations (μg/mL) unless otherwise noted: Gentamicin (Gm), 50; rifampicin (Rf), 50; theory Isopropyl β-d-1-thiogalactopyranoside (IPTG, 0.6 mM).

### 4.2. Genome Sequencing and Annotation

The genome of PSA.AH.01 was sequenced by the DNBSEQ platform (BGI, Shenzhen, China)and Nanopore platform (Oxford Nanopore, Oxford, UK). Sequence data were assembled using Falcon, version 0.3.0, and Celera Assembler, version 8.3, and analyzed using Glimmer, version 3.02. Then the protein sequence of the genome was blasted with the VFDB for gene annotation to find the potential virulence factors.

### 4.3. Construction of the Base-Editing System pdCas12a-BE

The vector backbone was obtained from pBmBE3 [3]. The dCas12a (dCpf1) and NLS (Nuclear Localization Signal of SV40) fusion sequences were amplified from the dCpf1-BE vector to replace the dCas9 region by ClonExpress II One Step Cloning Kit (Vazyme, Nanjing, China), resulting in pAP-dCas12a-NLS [31]. The NLS and APOBEC1 sequences were amplified and fused into the NLS-APOBEC1 sequence. The APOBEC1 region of pAP-dCas12a-NLS was replaced by the NLS-APOBEC1 sequence, resulting in pNLS-AP-dCas12a-NLS. Then, the two *Pst*I restriction sites were removed using the Fast Mutagenesis System (Transgen Biotech, Beijing, China). The crRNA-BsaI sequences were synthesized and inserted into the site between *Bsa*I and *Pst*I by T4 DNA Ligase (NEB, Beijing, China), resulting in pdCas12a-BE (Appendix A, Appendix A and Supplementary Notes 1).

### 4.4. Target Design Procedure

For the dCas9-BE3 system, finding the NGG structure in half of the target gene’s open reading frame is necessary. If the CAG/CAA (Gln) or CGA (Arg) codons were located at 20 bases upstream of the PAM site, then this region would be suitable as a target. For the dCas12a-BE system, finding the TTTN structure in half of the target gene CDS is necessary. If the CAG/CAA (Gln) or CGA (Arg) codons were located at 20 bases downstream of the TTTN structure, this region could be designed as a target site.

### 4.5. Knockout Vector Construction

For a single-site knockout, the target region’s sense and anti-sense strands were synthesized and annealed to form a dimer. For two-site knockout, the ‘Target1-sgRNA-Trc-Target2’ was assembled by overlapping PCR. Then the fragment was ligated into BsaI-digested *pBmBE3* or *pdCas12a-BE* vectors.

### 4.6. Construction of a Genome-Wide Mutant Library

For the dCas9-BE3 system, the 12-bp seed sequence of the sgRNA should only match with one site in the target genome, so the possibility of off-target editing is very low. If the CAG/CAA (Gln) or CGA (Arg) codons were located at 20 bases upstream of the PAM site, then this region would be suitable as a target. Hence, to generate loss-of-function mutations efficiently, sgRNA target sites were designed in the first half of the open reading frame, as close as possible to the start codons. According to those editing rules of Cas9-cytidine deaminase fusion in *Psa*, we used Python to screen out one or two suitable targets for each gene in the PSA.AH.01 genome. Twenty-five base pairs were added at the 5′ end of the target, which identifies with upstream *BsaI* in pBmBE3.

Similarly, twenty-five base pairs were added at the 3′ end of the target, which identifies with downstream *BsaI* in pBmBE3. The oligonucleotides (each oligonucleotide was 70 bp) were synthesized by array-based oligonucleotide pool synthesis. Then all oligonucleotides were amplified by PCR. The PCR products were purified and ligated into the BsaI-digested pBmBE3 vector by Gibson ligation, and the successfully ligated plasmids were transformed into *Psa* or *E. coli* DH5α chemically competent cells (Figure 6).

For the dCas12a-BE system, finding the TTTN structure in half of the target gene CDS is necessary. If the CAG/CAA (Gln) or CGA (Arg) codons were located at 20 bases downstream of the TTTN structure, this region could be designed as a target site.

### 4.7. Plasmid Transformations into Psa

The *Psa* strain was taken from an ultra-cold storage freezer (−80 °C) and cultured on King’s B medium (containing 50 μg mL^−1^ Rifampicin) at 25 °C for 48 h before usage. A single colony was inoculated in 5 mL of liquid KB and shaken at 200 rpm min^−1^ overnight at 25 °C. The bacterial culture (2 mL) was transferred to 50 mL of liquid KB and incubated in the same conditions until the OD_600_ reached 0.5. Cells were collected by centrifuging at 4 °C for 10 min at 10,000× *g* and resuspended with a 5 mL cold CaCl_2_ solution (20 mM). The supernatant was discarded, and the bacterial cells were resuspended with the buffer (0.7 mL of 20 mM CaCl_2_ and 0.3 mL of 50% glycerol) and then dispensed into each centrifuge tube with 100 μL. The competent cells were immediately frozen in liquid nitrogen for 5 min and stored at −80 °C (Supplementary Notes 2 and Appendix A). The pBmBE3 or pdCas12a-BE series plasmids were transformed into chemically competent cells. After 30 min of an ice bath and heat shock at 42 °C for 4 min, the recovered transformed *Psa* cells were added to 0.5 mL of liquid KB (0.6 mM IPTG) and cultured at 25 °C for 2 h at 180 rpm. Cells were inculcated on the KB solid medium (containing 50 μg/mL gentamicin and 0.6 mM IPTG) for 2 days at 28 °C. A single colony was selected, and the target gene was amplified and sequenced to identify the mutation events.

### 4.8. Pathogenicity Assays

*Psa* infection assays were based on previous procedures [41]. The axillary buds of *A. chinensis var. chinensis ‘Hongyang’* tissue culture plantlets were cut and grown on Murashige and Skoog rooting medium without antibiotics in plates. Plantlets were grown in a plant growth chamber with a temperature of 24 ± 1 °C and a photoperiod of 16 h light/8 h dark. Kiwifruit plantlets (4 weeks old) were infected using a plant flooding method [42]. *Psa* (containing empty vector) and mutants were grown in liquid LB (containing 50 μg/mL gentamicin) followed by overnight shaking and washing, and the cell density was adjusted to 1 × 10^7^ CFU/mL. Silwet L-77 (Coolaber, Beijing, China) as a surfactant was added to the *Psa* suspension at 0.0025% (vol/vol). Kiwifruit plantlets were fully submerged in *Psa* suspension for 5 min and grown in a plant growth chamber with a temperature of 16 °C with 16 h/light and 10 °C with an 8 h/dark cycle. Leaves were collected at 20 days postinoculation (dpi). Leaf samples were washed with ddH_2_O three times and ground with 1 mL of sterile 10 mM MgSO_4_. The leaf homogenate was stored at 4 °C overnight, and the supernatant was diluted 10-fold and plated as 10 μL droplets on the LB medium with 50 μg/mL Gm. After 2 days of incubation at 25 °C, the CFU per cm^2^ leaf was ascertained from dilutions.

In this paper, the screening of *Psa* pathogenic genes focused on the effector proteins of the T3SS in the sequenced strain PSA.AH.01. We referred to relevant literature regarding DC3000 and selected 10 effectors, namely, HopAO2, HopR1, HopAI1, AvrE1, HopM1, HopI1, AvrB4, HopQ1, HopZ3, and HopF2, which have been reported in the literature [19,38,40,43,44,45,46,47,48,49].

## Figures and Tables

**Figure 1 ijms-24-04597-f001:**
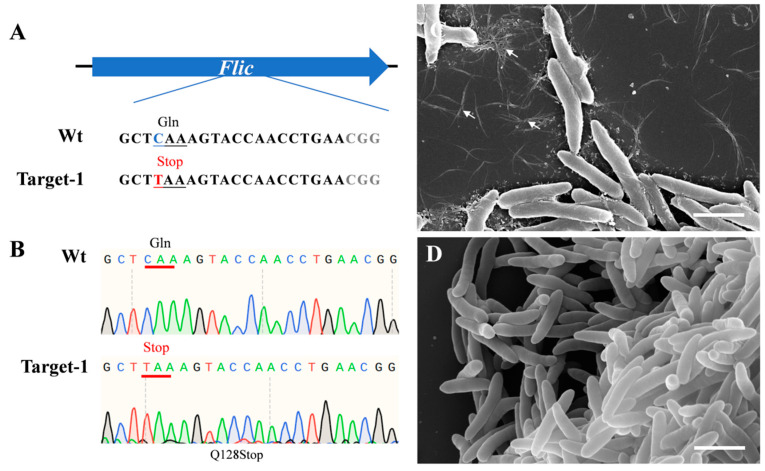
Induction of base conversion at *fliC* sites in *Psa* and flagella phenotype. (**A**) The designed mutation sites in the *fliC* gene. PAM motif (gray), target sites (red). (**B**) Sanger sequencing at base-editing site of Flic gene. (**C**) Flagella (arrows) of *Psa*; (**D**) no flagella of the Δ*fliC* mutant were observed. Bar = 2 μm.

**Figure 2 ijms-24-04597-f002:**
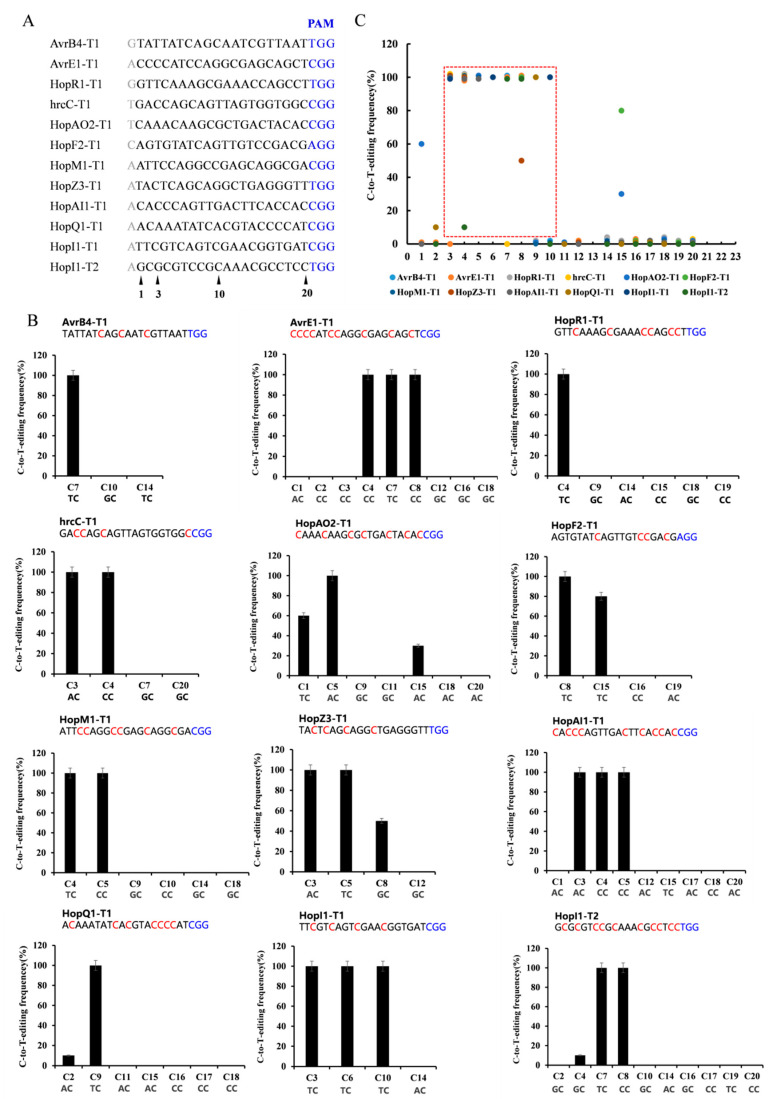
Base editing efficiency of dCas9-BE3 in *Psa*. (**A**) The editing window of 12 target-site sequences. The cytosines were counted with the base distal to the PAM (blue) setting as position. The major editing window is C3-C10 (red frame). (**B**) Summary of C-to-T-editing frequency at each cytosine of 12 target sites. (**C**) The C-to-T-editing frequency of TC, CC, AC, and GC structures in the editing box. The PAM is marked blue. The phenotype assay and Sanger sequencing were performed on 10 colonies of each group. The editing frequency was calculated by formula (edited colony/total colony). Similar to the following.

**Figure 3 ijms-24-04597-f003:**
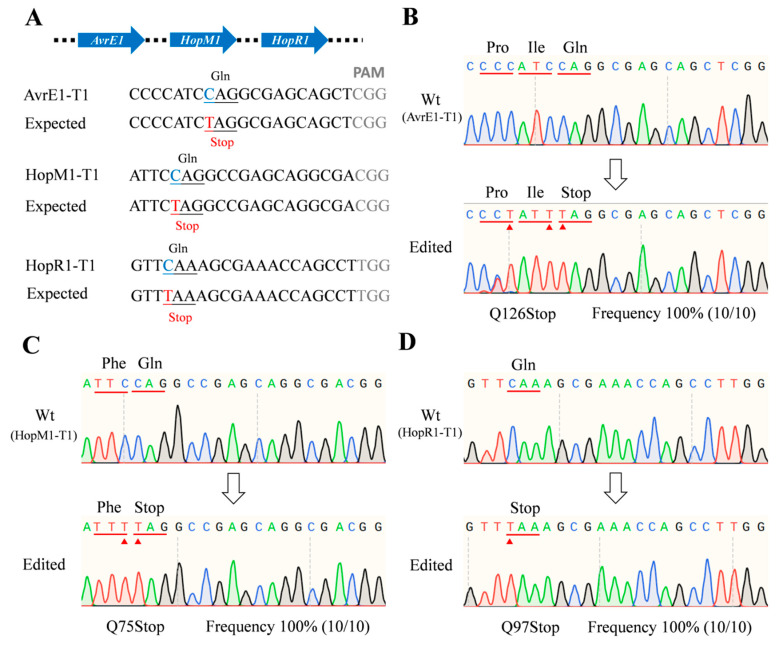
Muti-site knockout using the dCas9-BE3 system in *Psa*. (**A**) The designed mutation sites in the *avrE1*, *hopM1*, and *hopR1* genes. PAM motif (gray), target sites (blue), substituted bases (red). (**B**–**D**) Sanger sequencing of base-editing in the *avrE1, hopM1*, and *hopR1* gene sites. Substituted amino acids (red line). The Sanger sequencing was performed on 10 colonies of each group. The editing frequency was calculated by the formula (edited colony/total colony).

**Figure 4 ijms-24-04597-f004:**
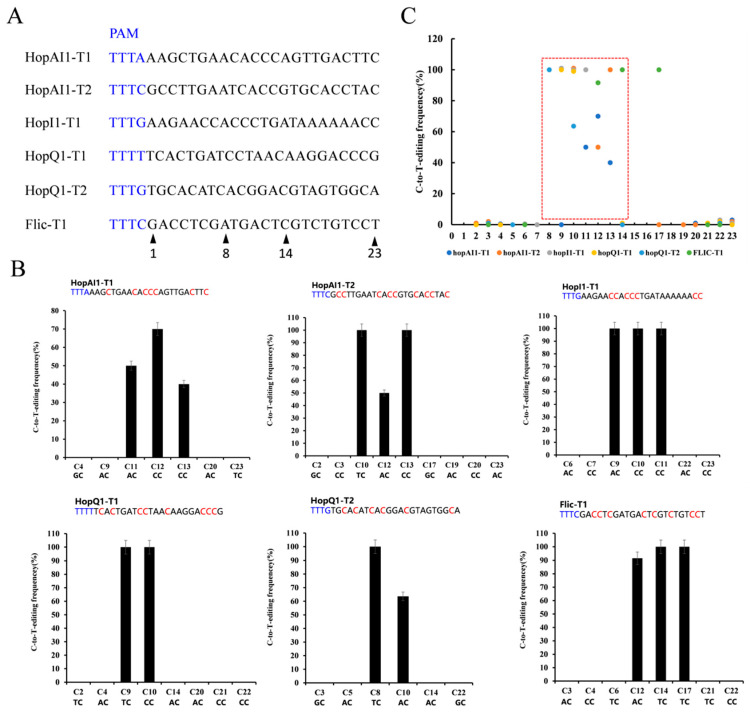
Editing rule of pdCas12a-BE in *Psa*. (**A**) The designed mutation sites. PAM motif (blue). (**B**) Determination of dCas12a-BE-induced base-editing frequency at every single cytosine in the indicated spacer region. The cytosines were counted with the base proximal to the PAM setting as position 1. (**C**) Summary of the base editing frequency at each cytosine in the spacer region for the six indicated crRNAs. These data show that the major editing window ranges from positions 8 to 14 in the spacer region (red frame).

**Figure 5 ijms-24-04597-f005:**
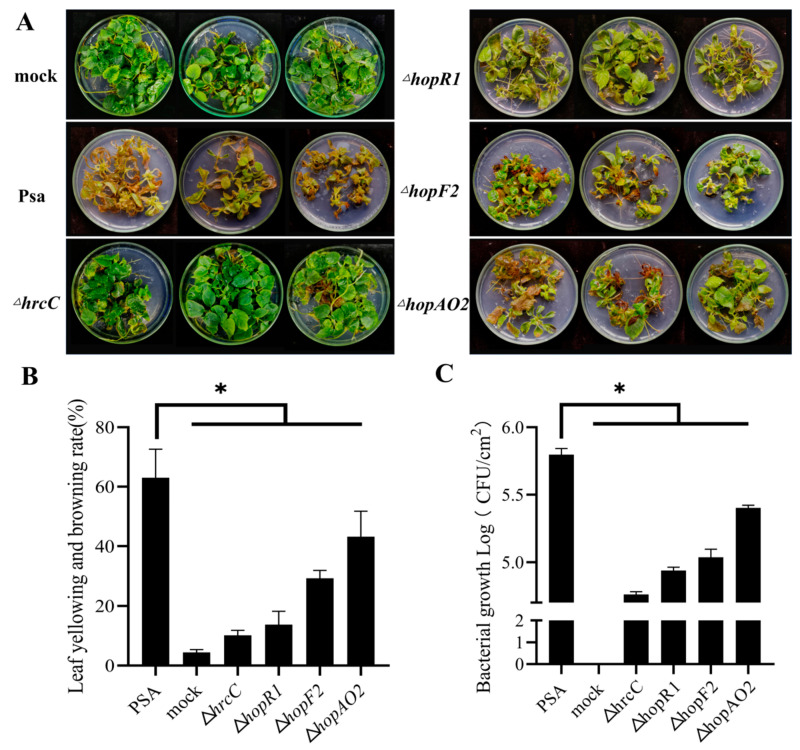
Phenotype of ‘hongyang’ plantlet after inoculation with *Psa* mutant strains. (**A**) Symptom of ‘hongyang’ plantlet after infection by Δ*hrcC*, Δ*hopR1,* Δ*hopF2*, and Δ*hopAO2* (20 days post-inoculation). (**B**) Leaf yellowing and browning rate of ‘hongyang’ plantlet after inoculation with *Psa* and mutant strains. Error bars represent standard error of the mean from three pseudobiological replicates. Asterisks indicate results of a two-tailed Student’s *t* test between the selected sample and wild-type *Psa*; * *p* < 0.05. (**C**) The bacterial growth of *Psa* and mutant strains in ‘hongyang’. Error bars represent standard error of the mean from three pseudobiological replicates. Asterisks indicate results of a two-tailed Student’s *t* test between the selected sample and wild-type *Psa*; * *p* < 0.05.

**Figure 6 ijms-24-04597-f006:**
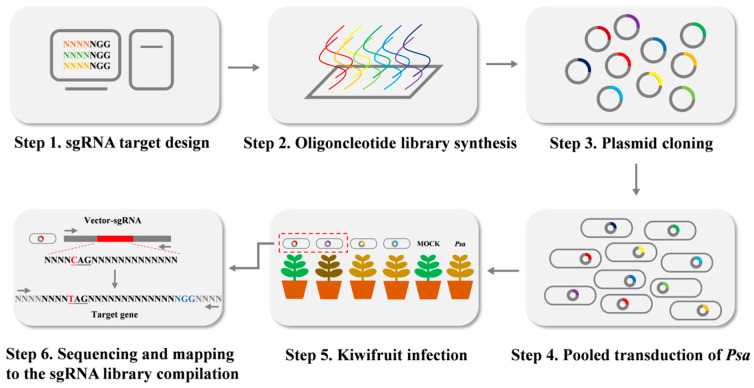
Schematic illustration of CRISPR/Cas9 sgRNA library construction in *Psa*.

## Data Availability

The datasets generated for this study can be found in the BioProject database of NCBI with accession PRJNA923731.

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
