# Peer review of "dCas9-BE3 and dCas12a-BE3 Systems Mediated Base Editing in Kiwifruit Canker Causal Agent Pseudomonas syringae pv. actinidiae"

_ijms, 2023, doi:10.3390/ijms24054597_

Round 1

Reviewer 1 Report

Liu et al investigated the utilization of CRISPR base editing tools (dCas9 and dCas12a) in Pseudomonas syringae pathogen. Authors demonstrated nicely precise edits and also generated pathogen mutant library. Manuscript is well designed and results are nicely presented. I have only two comments

What does 3 means BE3

It is not clear what does mean mutant efficiencies. It this related to total clones or related to total target site within gene. Please clarify in the text

Have authors done any analysis to determine off targeting 

Reviewer 2 Report

This manuscript described on the development of two kind of dCas system, construction of a genome-wide mutant library and identification of pathogenicity causing genes in kiwifruit canker causal agent Pseudomonas syringae pv. actinidiae.

 Although this manuscript is well written and experimental procedures are quite excellent, it is difficult to evaluate the results of the findings of pathogenicity causing genes, hopF2 and hopAO2. The authors should explain the functions of the genes in detail. In addition, it was difficult to understand the reasons for the candidate genes for pathogenicity assays.

 Major points:

1.    Is the kiwifruit canker causal agent Pseudomonas syringae pv. actinidiae deposited to authorized culture collection? If the authors did not deposit it, the strain should deposit to an authorized culture collection to allow a third-party try to check the reproducibility of these experiments.

2.    Normally, the first paragraph of “Discussion” describes the most appearing result(s) in the paper. Although the base editor (BS) system has been developed in Li et al. in 2016 (Komor et al., Nature, 2016 533) and the procedure has been applied in many bacteria, the authors described on the application of BS system for Psa in the first paragraph of “Discussion”.

3.    L263: CAG → TAG, CAA → TAA

Minor points:

1.    L30, L32 and the other parts: Please consider the significant digits for “76.88%” and “75.56%”.

2.    L154 and the other parts: Please describe name of bacterial species for “DH5α”.

Round 2

Reviewer 2 Report

The manuscript described on the development of CRISPR-based base editing in the plant disease causing agent Psuudomonas syringae pv. actinidiae.

 Although it was difficult to understand the availability of PSA.AH.1 by a third party, the manuscript was intensively improved. Description of stain number in CGMCC is desirable.

1)      L38: Please add the brief explanation for the function of the genes.

2)      L173: Please make “600” to subscript.

3)      “g” of “10,000 g” should be italics.

4)      L422: Pseudomonas syringae should be italics.
